# Explaining the Attributes of a Deep Learning Based Intrusion Detection System for Industrial Control Networks

**DOI:** 10.3390/s20143817

**Published:** 2020-07-08

**Authors:** Zhidong Wang, Yingxu Lai, Zenghui Liu, Jing Liu

**Affiliations:** 1College of Computer Science, Faculty of Information Technology, Beijing University of Technology, Beijing 100124, China; rikaaa0928@gmail.com (Z.W.); jingliu@bjut.edu.cn (J.L.); 2Automation Engineering Institute, Beijing Polytechnic, Beijing 100176, China; 100150@bpi.edu.cn

**Keywords:** deep learning, industrial control network, intrusion detection system, layer-wise relevance propagation

## Abstract

Intrusion detection is only the initial part of the security system for an industrial control system. Because of the criticality of the industrial control system, professionals still make the most important security decisions. Therefore, a simple intrusion alarm has a very limited role in the security system, and intrusion detection models based on deep learning struggle to provide more information because of the lack of explanation. This limits the application of deep learning methods to industrial control network intrusion detection. We analyzed the deep neural network (DNN) model and the interpretable classification model from the perspective of information, and clarified the correlation between the calculation process of the DNN model and the classification process. By comparing the normal samples with the abnormal samples, the abnormalities that occur during the calculation of the DNN model compared to the normal samples could be found. Based on this, a layer-wise relevance propagation method was designed to map the abnormalities in the calculation process to the abnormalities of attributes. At the same time, considering that the data set may already contain some useful information, we designed filtering rules for a kind of data set that can be obtained at a low cost, so that the calculation result is presented in a more accurate manner, which should help professionals lock and address intrusion threats more quickly.

## 1. Introduction

Industrial control systems (ICS) greatly increase the efficiency of industrial production by automating the control of industrial equipment and production processes. This makes the flow characteristics and importance of the industrial control network quite different from those of an ordinary network environment [1]. These differences result in special requirements by industrial control systems for intrusion detection systems. Cyberattacks pose a significant threat to industrial control systems, thus, a faster and more effective response to network intrusions is needed.

The response to complex cyberattacks is often determined by a professional team; professionals are irreplaceable in the security field [2]. Thus, much work still needs to be done after an intrusion detection system raises the alarm. This work is complicated but also needs to be done quickly. If an intrusion detection system (IDS) can provide more information, this would be very helpful for follow-up work.

The maturity of deep learning had led researchers to increasingly focus on its application to IDSs [3,4,5]. Compared with other algorithms, deep learning has many excellent features, such as a generalization ability and continuous learning ability [6]. These features greatly reduce the cost of manual analysis; deep learning can even perform better in some areas and can be more easily adapted to new changes or upgrades with constant feedback.

The biggest drawback of deep learning is that it is a “black box” model and lacks interpretability. Only the output information can be obtained directly; obtaining information from the model itself and its operation is difficult. Thus, a deep learning model has difficulty in ensuring that the IDS provides more intrusion information. A common solution to this problem is to enrich the model output or use interpretable models to approximate black box models, which increases the cost of manual analysis [7].

A possible solution is layer-wise relevance propagation (LRP) [8], which can be used to explain individual network decisions. This method has been successfully applied to explaining which pixels of an image are related to the classification decision [9], electroencephalogram (EEG) data analysis [10], and therapy prediction [11]. LRP splits a model in layers and assesses the relevance level of each layer in turn to achieve good results in the image field. Thus, LRP is a feasible method for increasing the interpretability of deep learning networks. Intrusion detection and image classification differ in that the latter focuses on the relevance between the input and result label, whereas the former focuses on the cause of the anomaly.

The deep learning model can learn knowledge from the data through the training process, which is used for classification for the classification model. Furthermore, the knowledge can be defined as classification bases of the classification process. In a rule-based classification model, each rule is a classification basis in the model. Therefore, we can consider classification bases as having the same nature as classification rules. We propose a method of analyzing the information and functions contained in the hidden layer of the deep learning model using a concept similar to LRP. The analysis results showed that the information contained in the hidden layer is closely related to the model classification basis and indicated the feasibility of the method from the perspective of information. The information contained in the hidden layer is reflected in the input by LRP, which can be used for further intrusion analysis after detection. We designed an experiment to verify the feasibility of the proposed method.

Using this method, a simple training set that does not need to contain too much information can be used to train an intrusion detection model, while still having the ability to quickly analyze the relevance of each input attributes on the results of each intrusion behavior. This greatly saves the analysis cost of the training set and the analysis cost of each intrusion during the detection phase.

Finally, this paper proposes a method that can obtain the relationship between the input difference and the output abnormality through layer-wise relevance calculation without modifying the structure and training of the existing deep neural network (DNN) model. Additionally, it proposes some methods that can improve the accuracy and readability of the results. Moreover, experiments were used to verify the effectiveness of the method in the ICS intrusion detection environment. Using the relevance calculation method can effectively improve the overall process efficiency of security threats in the ICS environment.

## 2. Related Works

The main purpose of this paper is to obtain a method to explain the attributes of the intrusion data in the industrial control intrusion detection environment through the analysis of deep learning models. The analysis and interpretation of the DNN model is the main content of the research, and the intrusion detection system in the ICS is the basis of the research.

Carcano et al. proposed an intrusion detection method by tracking the critical states. They believe that attacks always try to get the system out of a safe state and enter a critical state that can cause dangerous or harmful situations. Since the number of critical states is limited and all possible critical states can be obtained through analysis, the current threat state of the system can be evaluated by tracking the distance between the current system state and the critical state [12]. Pan et al. proposed an intrusion detection method based on a sequential pattern mining algorithm to process the sequences of critical system states, detecting disturbances and cyber-attacks in power systems [13]. Cheung et al. proposed three model-based detection methods for monitoring SCADA (Supervisory Control and Data Acquisition) networks; these methods monitor the system according to policies specified by valid sequences of system behaviors, where any sequence of behaviors outside the predefined specifications is regarded as an abnormal behavior [14].

The above research relies on manual summarization and analysis to transform human expert knowledge to rules that can be used by machines for intrusion detection. Such manual analysis is an expensive and arduous task, and the quality cannot be guaranteed. Under such conditions, machine learning (ML) methods that have excellent scalability and can automatically complete all or part of the learning task demonstrate great advantages. At present, there have been many related works using machine learning methods in intrusion detection of industrial control systems.

Alves et al. proposed a method about identifying the normal network pattern, attack signatures and man-in-the-middle attacks, and developed an Intrusion Prevention System (IPS) on programmable logic controllers (PLCs) with ML [15]. Feng et al. proposed an intrusion detection method using Long Short-Term Memory (LSTM) for ICS [16]. LSTM can learn the temporal context relationship between data and can be used to detect attack data sequences. Hadžiosmanovic et al. proposed a method using n-gram based algorithms to network traffic anomaly detection [17]. The test results show excellent profiles for true and false positives. Hasan et al. proposed a method using a constraint-oriented protocol definition to define normal and identify anomalies [18]. Li et al. proposed a method using a sparse auto-encoder-extreme learning machine intrusion detection model for the problem of intrusion detection accuracy. It uses deep learning autoencoder to combine the coefficient penalty and reconstruction loss of the encode layer to extract the features of high-dimensional data during the training model, and then uses the extreme learning machine to quickly and effectively classify the extracted features [19]. Ahmad et al. proposed a deep learning approach to implement an effective and enhanced IDS for securing industrial network [20]. Yang et al. proposed a convolutional neural network (CNN) based network intrusion detection system for SCADA networks to protect ICSs from both conventional and SCADA specific network-based attacks [21]. Chen et al. proposed an industrial control system intrusion detection method based on gate recurrent unit neural network to handle the problem that the intrusion detection method based on a traditional machine learning algorithm cannot effectively deal with massive, high-dimensional, time related network traffic data in industrial control system [22]. Shi et al. proposed an intrusion detection model based on correlation information entropy and CNN-BiLSTM. It combines feature selection based on correlation information entropy with fused deep learning algorithms, and thus, it can effectively remove noise redundancy, reduce computation, and improve detection accuracy [23].

The deep learning model has been widely used in the field of ICS intrusion detection and has certain advantages. Most of the research has focused on making the model adaptable to more needs and improving accuracy. However, obtaining additional attack clues through the interpretation of the model is also very important for the security system in ICS. In fact, there have been many related studies in various fields focused on improving the interpretability of the black box model at all levels.

Datta et al. introduced a set of Quantitative Input Influence (QII), which measures how much inputs influence the outputs of black box predictors, the output consists in the feature importance for outcome predictions [24]. Sundararajan et al. introduce an attribution method called Integrated Gradients (IG), which requires no modification to the original network [25]. Goldstein et al. introduce a method that extends classical PDP named Individual Conditional Expectation (ICE) to visualize the model approximated by a black box that helps in visualizing the average partial relationship between the outcome and some features [26]. Yosinski et al. introduce two tools for visualizing and interpreting DNNs and for understanding what computations DNNs perform at intermediate layers and which neurons are activated. They found that by analyzing the live activations and observing as they change in correspondence of different inputs can helps to explain the DNNs behavior [27]. Zeiler et al. introduced a method that backtracks the network computation to map the relationship between image patches and certain neural activations [28]. Simonyan et al. showed that Zeiler’s method can be interpreted as a sensitivity analysis of the network input/output relation [29]. Springenberg et al. introduced a new variant of the “deconvolution approach” to visualize the features learned by model [30]. Radford et al. presented the discovery that a single neuron unit of a DNN can perform a sentiment analysis after the training of the network, reaching the same level of performance of strong baselines [31].

Most of the relevant research on the interpretability of the model is carried out in the fields of image processing and word processing, and has achieved good results. However, in the ICS intrusion detection environment, the data dimension is smaller and the information contained is more important, thus, the requirement of accuracy is also higher. At the same time, compared to the image processing and word processing fields, the data scale in the ICS intrusion detection environment is smaller, the required model complexity is lower, and the network scale is also smaller. Therefore, using an approximate method derived in the ICS intrusion detection environment may obtain better accuracy.

## 3. Explaining the Method of Deep Learning Models

The proposed method is an extension of the traditional intrusion detection method. An intrusion detection model is required, and a negative sample for comparison is needed for relevance analysis. First, data are collected, and a deep learning classification model is trained for intrusion classification. To determine the best case for finding a comparison sample, the role of each hidden layer of the classification model in the information transfer needs to be analyzed. Once an intrusion is detected, a comparison sample can be obtained from a previous analysis, and the relevance between each input dimension and the resulting change can be calculated from the positive and comparison samples.

To increase the accuracy of the relevance calculation, a deeper learning model with a simpler structure and smaller size can be selected without significantly affecting the model accuracy. Compared with traditional networks, the normal traffic of an industrial control network shows obvious regularity [32], but some attack traffic will show randomness. The normal min–max feature scaling normalization method reduces the discrimination of data in the normal domain. We introduce a new data normalization method to improve the discrimination of some input data in our proposed method.

LRP is a widely used algorithm. The proposed method uses the same concept as LRP: backpropagation of the results from the output layer back to the input layer. It has shown excellent performance in many fields, but the job is to try to extract information from deep learning models and use the results to improve the process of intrusion detection for such important system as industrial control networks. We want to know why the work of extracting information from the model can be done by back propagation, combined with the nature of intrusion detection, and design a targeted back propagation calculation method based on the nature of intrusion detection, so as to help the intrusion detection process more efficiently and accurately. However, LRP is a mathematically derived method; thus, we analyzed the calculation process of the deep learning model from the perspective of information [33] and combined it with the characteristics of the intrusion detection process, that is, the classification process. Finally, a new feasible method is found through hypothesis and verification. According to our experiments, during the transmission process of the deep learning classification model, bases unrelated to the classification are discarded layer by layer, and the relevant bases are retained. Different degrees of discrimination are displayed at different levels. Selecting the hidden layer output with the highest degree of discrimination in this category as a comparison sample can highlight the role of relevant bases.

To calculate the relevance, we focused on the difference in results instead of the output value. Because an input sometimes makes a huge contribution to the desired output dimension but also makes a huge contribution to the unwanted output dimension, the relevance of static inputs is meaningless for intrusion basis analysis.

In addition, in the purpose of verifying the experimental results, the intensive classification and information-rich data set is used as the training set in the experiment, but the training set that satisfies the method only needs to meet the conditions that can distinguish the intrusion data from the normal data. Therefore, this method can also greatly reduce the cost of collecting training sets. At the same time, if more information contained in each class of the training set can be obtained (such as distinguishing command injection attacks and response injection attacks), the information can be rationally utilized to establish filtering rules to further streamline the calculation results or correct the calculation results.

Finally, a small experiment was designed to test the influence of the new normalization method on the training process. The results show that using this method can effectively solve the problem that needs to continuously reduce the learning rate to improve the accuracy of training.

### 3.1. Data Normalization

Industrial control devices generally are in a stable state during normal operation, and the industrial control network traffic also shows strong regularity. Programmable logic controller (PLC) registers have a wide range of values, but their read and write values are often concentrated in one or several small ranges for normal industrial network traffic. However, when an intrusion occurs—especially a naive malicious response injection (NMRI) or malicious parameter command injection (MPCI)—random or very large and very small data are directly used to attack the PLC. Denial of service (DoS) attacks also cause the time interval parameters to deviate significantly from normal levels. These cause the problem that the minimum and maximum values in the dataset tend to be closer to the theoretically feasible input range when min–max feature scaling normalization is used for model training or evaluation, which is much larger than the normal input range during normal operation. Finally, the data in the normal state are scaled to a very small range, and data discrimination is likely to be lost. However, other intrusions, such as the complex malicious response injection (CMRI), which is designed to appear to have normal process functionality, can be used to mask alterations to the process state by malicious command injection attacks. The model needs to distinguish CMRI from data in the normal state, but if the data discrimination in the normal state is lost, this causes a problem for the detection process called over-scaling.

The above problem can be solved with other normalization methods such as standard score normalization, but this method is designed for data that conform to a normal distribution, and controlling the data distribution in the training set is generally difficult. We propose a generalized normalization method for dealing with normal and abnormal labeled data and large differences in data ranges.

First, not all of the input dimensions have the over-scaling problem; thus, these need to be found and taken out. The remaining dimensions can be normalized by common methods. Second, for each potentially over-scaled dimension, the minimum and maximum values of the normal labeled data and full dataset should be designated as norm_min, norm_max, full_min, and full_max as in Figure 1. Third, the input data should be split into two dimensions. The values in the first dimension that fall into the normal range are normalized with norm_min and norm_max as the minimum and maximum values. In the second dimension, values outside the normal range are normalized with full_min and full_max−(norm_max−norm_min) as the minimum and maximum values as Figure 2. Finally, the two new dimensions are normalized with new values and ranges to preserve the meaning of the number itself and alleviate the over-scaling problem. Composed x is input, norm is the first dimension, abnorm is the second dimension, nmin, nmax, fmin, fmax are the short names of norm_min, norm_max, full_min, full_max, respectively. Additionally, min() is the minimum function, max() is the maximum function. The normalization method can be presented as below:(1)$$\{\begin{array}{c} norm=\min\left(\max\left(\frac{x-nmin}{nmax-nmin},0\right),1\right)\\ abnorm=\frac{\min\left(x,nmin\right)-fmin+\max\left(x-nmax,0\right)}{fmax-nmax+nmin-fmin} \end{array}$$

### 3.2. Hidden Layer Analysis

The deep learning network is a black box from the perspective of interpretability but not from the perspective of model parameters. The value of each parameter in the model is known, and we can observe the detailed process of each calculation, but we lack a way to understand the content. Despite this, many researchers have tried to analyze deep learning models from various perspectives [34,35]. Although there is still no complete explanation for deep learning models, there are many theories that explain deep learning of certain processes or performances. Reverse engineering can be used to approximate a black box model with interpretable models. Interpretable models should be both able to mimic the behavior of a black box and also be understandable by humans. Alternatively, an interpretable local model can be built for a specific input instance and model, or a visual or textual representation can be used to understand a specific property of the black box model or its predictions. For example, LRP can indicate the relevance between the input and output, and the influence function can be used to calculate the influence of a training sample.

The proposed method was inspired by research on analyzing deep neural networks (DNNs) from the perspective of information [36]. Related research has shown that any DNN can be quantified by the mutual information between the layers and the input and output variables. Layered neural networks form a Markov chain of successive representations of the input layer. If H(X) is the entropy, I(X; Y) can be used to represent the mutual information of X and Y. According to the data processing inequality (DPI), for the input X and desired output Y, the output of each hidden layer $T_{i}$ and model’s true output $Y^{\prime }$ satisfy the following DPI chains as Equations (2) and (3):(2)$$I\left(X;\;Y\right)\geq I\left(T_{1};\;Y\right)\geq \ldots \;\geq I\left(T_{i};\;Y\right)\geq I\left(Y^{\prime };\;Y\right)$$
(3)$$H\left(X\right)\geq I\left(X;T_{1}\right)\geq \ldots \;\geq I\left(X;\;T_{i}\right)\geq I\left(X;\;Y^{\prime }\right)$$

This theory was used by Shwartz-Ziv and Tishby to calculate the theoretical optimal information limits of a DNN and obtain finite sample generalization bounds. We used this theory to improve the calculation of the relevance for the classification deep learning model. In a real intrusion detection and classification environment, the uncertainty of the tag data is much smaller than the input data. This means that $H\left(X\right)>I\left(X;Y^{\prime }\right)$ holds based on the above set of inequalities. In fact, the experimental results of A and B can clearly confirm this conclusion. At the same time, from their experimental results, it can be seen that in all experiments, the mutual information of the input data and the output of each hidden layer is decremented layer by layer. The conclusion could be described as Equation (4):(4)$$H\left(X\right)>I\left(X;T_{1}\right)>\ldots >I\left(X;\;T_{i}\right)>I\left(X;\;Y^{\prime }\right)$$

The feature of information changing layer by layer shows that the classification process of deep learning models is completed in steps. The decision tree model is easy to explain, and has a stepwise and directly related classification process and calculation process. To clarify whether and how the calculation process of the DNN model is related to the classification process, this paper divides the deep learning model and decision tree model from the calculation process and the classification process, and a comparative analysis was conducted at each level.

A decision tree consists of three types of nodes: decision nodes, chance nodes, and end nodes. Decision nodes represent choices based on attributes, chance nodes represent the expectations of non-end nodes after a decision, and end nodes represent the final results. For example, when $d_{i}$ stands for the decisions, $c_{i}$ stands for the chances after $d_{i}$, $d_{i,j}$ stands for the following decisions after $c_{i}$, $e_{i}$ stands for the ends, a decision tree model could be described with these variables as Figure 3.

The biggest advantage of the decision tree model is that each calculation step corresponds to an easy-to-understand classification step. Suppose that in the intrusion detection environment, the classification model shown in the figure is obtained after training, $e_{1}$ is a normal label, and the rest are attack labels. After an intrusion occurs, not only can the attack tags of the intrusion behavior be obtained, but also the key attribute that used by the model to make the key decisions during the classification process can be obtained through tracking the calculation process. For example, the main reason that causes the input data to be classified as an $e_{2}$ attack is the abnormality of related attributes in $d_{1,1}$ and $d_{1,2}$.

The logical decision process in the decision tree is defined as the function $f_{d}^{c}\left(x\right)$, which means that when the input *x* satisfies the condition d, c is returned, c contains the data and status after decision. Then the calculation process for the data of type $e_{2}$ can be described as $e_{2}=f_{d_{1,2}}^{e_{2}}\left(f_{d_{1}}^{c_{1}}\left(x\right)\right)$, where $c_{1}=f_{d_{1}}^{c_{1}}\left(x\right)$ is the intermediate result. In a real environment, after each decision, the information obtained will be more certain, that is, the uncertainty of the data will decrease as the discrimination process proceeds. According to the definition of mutual information, the mutual information relationship between the output data (data in chance nodes $c_{i}$ and end nodes $e_{i}$) and the input data x in the decision tree model is shown in Equation (5):(5)$$H\left(x\right)>I\left(x;c_{i}\right)>\ldots >I\left(x;\;c_{i\ldots j}\right)>I\left(x;\;e_{i}\right)$$

Comparing Equation (5) with Equation (3), it can be found that the change pattern of the mutual information of the DNN model in the layered calculation process is the same as that of the decision tree model. The mutual information change in the decision model is directly caused by the decision process. Therefore, it can be asserted that in the hierarchical computing of the DNN model in the real environment, there are similar characteristics to the decision tree. That is, the calculation process of each layer is related to the stepwise decision process. This gradual reduction of information can be understood as the model gradually discards information that is not related to the result during the calculation process, and retains and summarizes the information related to the result as the result.

If a DNN model equivalent to a decision tree model is trained, although one cannot directly understand the parameters and calculation process of the DNN model, the rules in the decision tree model still exist in the parameters of the DNN model in some way and expressed in layers. In this paper, the rules existing in the DNN model are defined as the classification basis. The same as the decision tree model, for a certain classification decision, the combination of the classification basis involved in the decision can be expressed as the overall-basis of the overall decision, and the classification basis constituting the overall-basis is defined as the stepwise-basis. The input and output of the DNN model are the embodiment of the role of the overall-basis, and the output of the hidden layer of the DNN model is the embodiment of one or some stepwise-basis. The stepwise-bases are key factors in explaining the decision behavior of the model.

Under this premise, in the intrusion detection environment, by comparing the calculation process of abnormal data with the calculation process of normal data, we can find the key decision-making layer (KDL) that causes the decision to change. The key decision-making layer reflects the role of key stepwise-basis. Then, the abnormality of the calculation process found in the comparison could be mapped to the abnormality of the corresponding attribute and the attributes of a DNN based Intrusion Detection System could be explained.

In the following sections, this paper designs an experiment to verify our assertions, and designs a relevance calculation method that maps the abnormalities in the calculation process to the input attributes according to the correlation.

### 3.3. Relevance Calculation

Anomalies are caused by differences. In the DNN classification model in an intrusion detection environment, for an anomaly sample and a comparison sample, the differences in input, output, and output of each hidden layer are related to the anomaly in varying degrees. Meanwhile, the difference in output directly represents the existence of anomaly, while only part of the difference in input is related to anomaly.

The relevance calculation method in this paper has a similar structure to the LRP method. Both methods use hidden layers for layer-by-layer relevance transfer. The difference is that the content of the transfer is different. The LRP method mainly focuses on the relevance between the input value and the output value, and the method in this paper needs to map the correlation between the output abnormality and the output abnormality.

In the DNN model, the calculation process of a specific hidden layer can be defined as y=f (wx+b), where x is the input, y is the output, f (x) is the activation function, and w and b are the parameters. When x is a multi-dimensional input, let $x_{i}$ represent the value of each dimension, and $w_{i}$ is the parameter corresponding to each dimension. The input differential $dx_{i}$ and the output differential dy can be obtained simply. According to the definition of partial derivatives, the partial derivative $p_{i}$ can be used as an important reference when analyzing the influence of each $dx_{i}$ on dy. It can be approximated that the relevance between $dx_{i}$ and dy is greater when the product of $dx_{i}$ and $p_{i}$ is greater. The above relationship is only established when $dx_{i}$ and dy are approximate to 0, but in an intrusion detection environment, it is often impossible to find comparative samples with similar values. To ensure that the value relationship of relevance is always established, some additional processing needs to be done for the case where the differential is large. As an example, for the common activation function tanh, when the differential is large, a method for repairing the numerical relationship is proposed as Figure 4.

Let tanh(x) be the activation function and A, B, C, and D be the samples in the dataset. (A, B) is a set of the target sample and comparison sample, $x_{A}$ is the value of sample A, $a_{A}$ is the activation of sample A, and $g_{max}$ is the maximum value of the derivative function of tanh(x). Make $g_{A}=\tanh{\left(x_{A}\right)}^{\prime }$, and $M_{\max\left(g\right)}$ is the sample that takes the maximum value of the partial derivative function. Make $d_{\left(A,B\right)}=d_{B}-d_{A}$. $R_{\left(A,B\right)}$ is the relevance value, and $R_{\left(A,B\right)}\propto d_{\left(A,B\right)}\times p$. p is the partial derivative, default set as $average\left(g_{A},\;g_{B}\right)$. For the example in Figure 2, $g_{A}=g_{D}=g_{1}$, $g_{B}=g_{c}=g_{2}$, and $g_{3}=average\left(g_{1},\;g_{2}\right)$.

For sets (A, B) and (A, $M_{\max\left(g\right)}$), have $d_{\left(A,B\right)}<d_{\left(A,M\right)}$ and $average\left(g_{A},\;g_{B}\right)<average\left(g_{A},\;g_{M}\right)$, thus, $R_{\left(A,B\right)}<R_{\left(A,M\right)}$. In this case, using the average value as p can correctly reflect the actual numerical relationship for the relevance from (A, B) and (A, $M_{\max\left(g\right)}$). However, for sets (B, C) and (B, D) with $d_{\left(B,C\right)}<d_{\left(B,D\right)}$ and $g_{2}>g_{3}$, the numerical relationship between $R_{\left(B,C\right)}$ and $R_{\left(B,D\right)}$ depends on the values in the actual calculation, but $R_{\left(B,C\right)}<R_{\left(B,D\right)}$ should always be true. A more detailed analysis shows that, when A and B are on the opposite side of $M_{\max\left(g\right)}$, for $R_{\left(A,B\right)}\;\propto d_{\left(A,B\right)}\times p$, p should be at least equal to $g_{max}$ to ensure that the value of R can be correctly assigned.

For single-layer models, let x be the input sample, $x_{d}$ be each input dimension of sample x, V be the dimensionality of sample x, $R_{d}$ be the relevance of each dimension, and D be the difference of classification output f(x). The abnormal relevance transfer method maps the output anomalies in each layer of calculation to each input dimension as in Equation (6):(6)$$D\approx \sum_{d=1}^{V}R_{d}$$

In the classification model, the relevance calculation method not only focuses on the increasing in the output of abnormal dimension, but also the decreasing in the output of normal dimension, making the results more accurate.

In the DNN model, the multi-layer computing structure can be split into multiple single-layer structures for continuous transfer. Let l be the number of layers, $n_{l,i}$ be the neuron i in layer l, $d_{l,i}$ be the difference of the output of the abnormal sample and the comparison sample at $n_{l,i}$, $p_{l,ij}$ be the partial derivative obtained by the above method, $R_{l,i}$ be the relevance of $n_{l,i}$, and $R_{l,i\leftarrow j}$ be the relevance transferred from $n_{l+1,j}$ to $n_{l,i}$ as in Figure 5. The relevance transfer calculation method can be described by Equation (7):(7)$$\{\begin{array}{c} R_{l,i}=\sum_{j=0}^{len\left(layer\;l+1\right)}\;R_{l,i\leftarrow j}\\ R_{l,i\leftarrow j}=R_{l+1,j\;\frac{d_{l,i}\;p_{l,ij}}{\sum_{h\;in\;len\left(layer\;l\right)}d_{l,h}\;p_{l,hj}}} \end{array}$$

Finally, the relevance calculation method can be described by Equation (8):(8)$$R_{l,i}=\overset{len\left(layer\;l+1\right)}{\underset{j=0}{\sum }}\;\frac{d_{l,i}\;g_{l,ij}}{\sum_{h\;in\;len\left(layer\;l\right)}d_{l,h}\;g_{l,hj}}R_{l+1,j}$$

The last problem is the determination of comparison samples. Since the correlation calculation method is an approximation-based method, the smaller the difference, the closer to the real result. A method for determining comparison samples that minimizes the difference and can maximize the impact of key decisions is required. According to the previous analysis, the hidden layer (KDL) where the key decision-making functions can be obtained. Using the output of KDL to find the comparison sample can maximize the method requirements.

In this section, the example DNN model use fully-connected layer with tanh(x). In theory, models of various structures can be supported by this method, and various activation functions can also be used. In actual applications, other structures and activation functions also need to be analyzed to obtain a method that can keep the numerical relationship during relevance calculation. According to the actual design, this method will bring different levels of calculation error. If sigmoid(x) is used, the situation is very similar to tanh(x), and the situation is different when using relu(x); when using Convolutional Neural Networks (CNN), the situation is more complicated.

### 3.4. Summary

The described method consists of two parts: (1) training and hidden layer analysis of the model and (2) intrusion detection and relevance calculation. The first part only needs to be executed once during the model creation and model update, and the second part is the formal execution process. In brief, the first part is as follows:Train a deep learning classification model with the improved normalization method.Obtain the KDL of each classification label by cluster-based analysis.In brief, the second part is as follows:Use the classification model to evaluate the input.If an intrusion is detected, obtain the KDL of the abnormal sample with the classification label.Find the comparison sample with the output of KDL.Calculate partial derivative *p* and obtain the relevance of each input with the proposed method.

In addition, this method will take longer to find a comparison sample when facing a larger data set, but the time complexity is o(n), and the time of single calculation is also very short, which will not cause excessive time overhead. In the relevance calculation method, intermediate results such as partial derivative numerical calculation can be cached to improve the performance of the method. Since some approximation methods are used in the derivation of the formula, the calculation error is inevitable, thus, the calculation error will be greater when the model is larger. However, it can be found through experiments that this method can complete the work well in a small-scale model under the ICS environment.

## 4. Experiments and Results

There were three parts to our experiments: the first part focused on the influence of the new normalization method, the second part focused on the hidden layer analysis, and the last part focused on the relevance calculation method. For the first part, an experiment was designed for the testing of the new normalization methods. For the second part, we used cluster analysis for the deep learning classification model to verify whether the assertion derived from the DPI chain was established in the deep learning intrusion detection model for industrial control. Thus, we derived a definition of the KDL and method of finding the KDL. For the third part, we applied the proposed method to an industrial control intrusion detection dataset for verification. The results showed that the proposed method can effectively help an analyst diagnose the intrusion details more quickly and play an important role in the IDS. In addition, the experimental code has been open sourced to codeocean (see Appendix A).

### 4.1. Testing of The Normalization Method

To verify the improvement that the normalization method can bring, an experiment was designed on the influence of the normalization method on the training process.

#### 4.1.1. Dataset for the Experiment

The dataset used in the experiment was a gas pipeline dataset in Industrial Control System Traffic Datasets for Intrusion Detection Research from Morris and Gao [37]. The data sets were captured using a network data logger, which monitored and stored MODBUS traffic from a RS-232 connection. The gas pipeline system includes a small airtight pipeline connected to a compressor, a pressure meter, and a solenoid-controlled relief valve. The pipeline system attempts to maintain the air pressure in the pipeline using a proportional integral derivative (PID) control scheme.

The dataset contains normal and abnormal data, and the abnormal data were divided into seven categories according to the type of intrusion. This experiment focuses on the training process, thus, a detailed introduction to the data set is placed in the next experiment.

#### 4.1.2. Details of the Experiment

Among the training parameters of deep learning models, the learning rate is an important parameter. Too large or too small a learning rate will cause the model to fail to converge properly or quickly. With other parameters being the same, the performance of the new normalization method and the traditional normalization method under several learning rates are compared. This can reflect the role of the new normalization method in the training process.

After analyzing the data, it was found that two of the 26 attributes of the dataset needed to be split, that is, the new normalization method generated a new dataset of 28 attributes. The network scale was set to [28/26, 64, 32, 16, 16, 8], where 28/26 is the input dimension and 8 is the output dimension; the hidden layer activation function is tanh; the optimization method is a Momentum Optimizer, and the momentum is set to 0.9; the batch size is set to 128; the experiment runs for 500 epochs. Three learning rates of 0.01, 0.005, and 0.001 are used in the experiment.

#### 4.1.3. Results of the Experiment

Deep learning models have a certain degree of randomness, and each training will not be exactly the same; there will even be large differences, thus, the results will give a range of values that often appear in multiple trainings instead of an exact number.

When the learning rate is set to 0.01, the final accuracy of both methods falls between 43–50%. When the learning rate is set to 0.005, the accuracy rate using the traditional normalization method falls between 45–55%, while the accuracy rate using the new normalization method falls between 55–63%. When the learning rate is set to 0.001, the accuracy using the traditional normalization method falls between 92–94%, while the accuracy using the new normalization method falls between 93–95%.

In addition, when dynamic (gradually decreasing) learning rate is used for training, using the new normalization method can easily train the accuracy to more than 95%, while using the traditional normalization method is much more difficult.

It can be seen from the experimental results that the new normalization method can bring a certain improvement to the training process. At the same time, it is more important that the new normalization method uses some prior knowledge in the intrusion detection data set, making the results of the correlation calculation more readable.

### 4.2. Hidden Layer Analysis

To verify whether the aforementioned assertion is true in the ICS intrusion detection environment, this paper designs related verification experiments. In the previous assertion, it is believed that the calculation process of the DNN model is related to the stepwise classification process. If the two are indeed related, it means that when a real category A has subclasses, A is determined by sets of different stepwise-bases. Due to the difference in the order in which stepwise-bases are executed, subclass data that also belong to A may show a large difference in the output of the hidden layer. These differences do not exist before the key stepwise-basis that caused the difference is executed, and after the stepwise-basis is executed, the difference will be revealed and finally disappear after the subclass of A is combined into A.

Through the previous analysis, with the layer-wise calculation of the DNN model, the uncertainty of the hidden layer output is gradually reduced, and the reduction of uncertainty is caused by the aggregation of data. The differences generated in the process of data aggregation are the research object of this paper, and the clustering method can effectively find the differences in the aggregated data.

#### 4.2.1. Dataset for Analysis

The same dataset as before was used. What follows are detailed descriptions of the dataset.

Naive malicious response injection (NMRI) attacks leverage the ability to inject or alter response packets in a network. However, they lack the ability to obtain information about the underlying process being monitored and controlled.

Complex malicious response injection (CMRI) attacks attempt to mask the actual state of the physical process and negatively affect feedback control loops. They are more sophisticated than NMRI attacks because they require greater in-depth understanding of the targeted system.

Malicious state command injection (MSCI) attacks change the state of the process control system to drive the system from a safe state to a critical state by sending malicious commands to remote field devices.

Malicious parameter command injection (MPCI) attacks alter programmable logic controller (PLC) field device setpoints.

Malicious function code command injection (MFCI) attacks use built-in protocol functions in a manner different from what was intended.

Denial-of-service (DoS) attacks target communications links and system programs in an attempt to exhaust resources.

Reconnaissance attacks gather SCADA system information, map the network architecture, and identify device characteristics (e.g., manufacturer, model number, supported network protocols, device address, and device memory map).

Generally, each type of attack corresponds to at least one stepwise-basis, but there is no way to know the number and content of these bases, which poses problems for the analysis of the correlation between the calculation process and the stepwise-bases. Therefore, in the experiment, all the abnormal classes in the data set are merged into one abnormal class, and the overall-basis of each original class is regarded as the stepwise-bases of the new abnormal class.

According to the relationship between the calculation process and the classification process of the DNN model described in the previous assertion, the aggregation of data in the hidden layer is related to the stepwise-bases, and the stepwise-bases are composed of the overall-bases of seven types of attacks. In this experiment, although only one binary classification model can be obtained, it should be possible to obtain clustering results related to seven types of attacks by performing cluster analysis on the output of the hidden layer. At the same time, due to the different execution order of the decision and the final combination, this correlation changes in trend of high-low or low-high-low.

It should be noted that because the data distribution of intrusion detection is very uneven, and since the ideal model cannot be obtained in actual scenarios, the analysis results may not be numerically perfect.

#### 4.2.2. Details of Analysis

The experiment consisted of four steps:A new intrusion detection classification model was trained with the two-class dataset and the improved data normalization method.The model was used to evaluate the correctly classified samples and record the output of all hidden layers during the evaluation process for analysis.The output of each hidden layer was clustered with a clustering algorithm.The clustering results were analyzed to determine whether they were in line with expectations.

Because of the large amount of data, many clustering algorithms with high complexity could not be successfully run. Only high-performance clustering algorithms such as k-means [38] and DBSCAN (Density-Based Spatial Clustering of Applications with Noise) [39] could be used in the experiments. K-means performs better with data that conform to normal distribution, thus, DBSCAN was used as the clustering algorithm in the experiment. A simple optimization method was adopted to adjust the parameters so that the number of classifications of DBSCAN would be equal to the number of targets and the noise samples are filtered in the analysis. Further experiments showed different clustering methods had little effect on the results. At the same time, considering that some classes may be determined by multiple atomic classifications, that is, these classes can be divided into more detailed subclasses, and these sub-classes may be clustered independently of each other in the hidden layer. The data set used has classes that meet this characteristic, and the subclasses is expected to be observed from the output of hidden layer, thus, the number of clustering targets should be more than the number of original labels, then, the subclasses belonging to the same class are combined. At the same time, to ensure that the subclasses obtained from the cluster really come from the corresponding classes, further analyzation and verification of the results is required.

To analyze the results, each multi-class cluster and the two-class model were combined into an independent multi-class intrusion detection classification model. By analyzing the precision and recall of each class by the combined model, we could see the correlation between the original results and clustering results for the data of each class. If TP is the true positive rate, FP is the false positive rate, and FN is the false negative rate, then precision=TP/(TP+FP) and recall=TP/(TP+FN). To ensure that the subclasses obtained from the cluster really come from the corresponding classes, the clustering labels were merged with the precision-prioritized rules to correspond to the original labels.

In addition, because the subject of the experiment is not the accuracy of the model, not much effort is spent on optimizing the model and the training process. The parameters of the model and the training process in the experiment only take some necessary changes based on the previous experiments.

#### 4.2.3. Results of Analysis

As described above, the dataset was used to train the deep learning model for intrusion detection. Then, the data generated by the sample in the calculation process were clustered. The precision and recall results are presented in Table 1, Table 2, and Figure 6. The calculation result can be used to measure the correlation between the clustering result and the original classification result.

The correlation results of the data of normal class show a low-high trend, mainly because in the early stage of the calculation process, some decisions for identifying abnormalities have not been executed, resulting in small difference between the normal data and some abnormal data.

The correlation results of the data of CMRI, MSCI, MPCI, and Reconnaissance classes show a high-low trend. There are two main reasons: 1. the data are highly abnormal and the data in the data set are not evenly distributed, thus, the data show difference from other data at the input stage. 2. Stepwise-bases related to these categories are executed in the first layer of calculation.

The correlation results of the data of remaining classes show a low-high-low trend, that is, the stepwise-bases related to these categories are executed in the middle of the calculation. According to the previous definition, KDL is the hidden layer that makes decisive decisions, that is, the first hidden layer with high correlation value is KDL.

### 4.3. Relevance Calculation

In this experiment, the proposed method was implemented. An intrusion detection model was trained to calculate the relevance of each data point in the dataset. In the calculation phase, the worst case that may actually occur was simulated, assuming that the specific information of each attack category in the data set is unknown, and then the calculation result of each intrusion category was compared with the actual intrusion information. The results are presented below.

#### 4.3.1. Dataset for Relevance Calculation

The same dataset as before was used. The applicability of the inferences was verified. With more original classes, the atomic bases were merged to a lower degree. This was more conducive to finding the MDH and improving the calculation accuracy. Thus, the original eight-class dataset was used.

#### 4.3.2. Details of Relevance Calculation

The experiment consisted of five steps:A new intrusion detection classification model was trained with the eight-class dataset and the improved data normalization method.The KDL was found for each class.The calculation method was used to analyze correctly classified samples.The test results were visually displayed by classification label and compared with the actual classification bases to check whether they were in line with expectations.

We drew the calculation results by classification label in the form of violin plots and scatter plots to see the distribution of intrusion factors in this class. Considering the expressiveness of the results, violin plot is used to show the calculation results of the classes with more samples and more scattered results, and a scatter plot was used to show the results of other classes. We then analyzed each intrusion type by comparing the calculated distribution of intrusion factors with the analysis results to see whether the algorithm can help analysts quickly narrow the scope of investigation and locate the key problem.

#### 4.3.3. Results of Relevance Calculation

First, we analyzed the model’s KDL. As in the previous experiment, we clustered each hidden layer output of all correctly classified anomaly samples and analyzed the distribution of clustering labels in the actual label. The results are presented in Table 3 and Table 4.

After the MDH was obtained for each anomaly class, the relevance between the input and resulting change in each sample was calculated with the proposed method. Figure 7 displays the normalized calculation results as scatter plots or violin plots.

For supervisory control and data acquisition (SCADA) systems of gas pipelines, NMRI attacks mainly use various strategies to modify the response value of the measurement. This causes abnormal pump and solenoid parameters and interferes with the normal operation of the system. The calculation results showed that the measurement, pump, and solenoid were detected with high confidence. The time and control_mode values were also calculated to have relevance but lower confidence or less detection. Combined with the new normalization method, a large number of attacks was found to fall within the abnormal data range of measurement.

CMRI attacks are designed to appear to have normal process functionality. These attacks can be used to mask alterations to the process state perpetrated by malicious command injection attacks, thus, they are more difficult to detect because they project a state of normalcy. A CMRI attack replays normal behavior or simulates abnormal behavior with abnormal frequency (time). This causes abnormal control parameters (control_mode, pump, and solenoid) and affects the normal operation of the system. The pertinence of CMRI makes the corresponding resp_read_fun also obtain a large negative relevance value. Combined with the new normalization method, the attack range of CMRI was found to fall within the normal range (measurement: n).

MSCI attacks may involve a single injected command or multiple injected commands. In the case of a gas pipeline system, an MSCI attack tampers with the control mode and turns on the compressor or pump to change the pressure in the pipeline. The calculation results showed that the control-related attributes all had different degrees of correlation, and the abnormal time caused by the injection attack was also calculated to be a small value. Thus, it was not a major factor. The corresponding resp_read_fun of the attack also obtained a large negative value.

MPCI attacks alter the set-point of the PLC field device through abnormal command_memory and command_memory_count, which causes abnormal control parameters and interferes with the normal operation of the system. It can be seen from the results that the parameters directly attacked by the attack and the abnormal control parameters caused by the attack are all detected, because the method does not have the ability to map the causal relationship of the parameters. Even so, this method still can reflect the characteristics of the attack well.

MFCI attacks use a malicious sub_function to interfere with the normal operation of the device, some of which cause the device to enter a non-working state. This causes abnormal time and control parameters and interferes with the normal operation of the system.

DoS attacks use a non-addressed slave address to continually transmit random data to random destination addresses to try to exhaust resources. The characteristics of the DoS attack data are highly random and the calculation results for this classification were also very messy.

Reconnaissance attacks collect the following information through traversal scans: device address, device supported function code, device identification, the memory map of MODBUS coils, discrete inputs, holding registers, and input registers. Because such attacks contain a large number of unrelated features, the results of the calculations showed clutter. Thus, it was impossible to determine whether the analysis results of such attacks were correct before further detailed analysis.

The above experiments and analysis showed that, other than DoS, the proposed method could accurately find the attributes related to an attack. The DoS attack was difficult to analyze at the attribute level but could easily be confirmed by other methods. Thus, this method can be used as an additional analysis tool for deep learning-based intrusion detection models to rapidly diagnose intrusion details even without any knowledge of the meaning of each intrusion class.

#### 4.3.4. A Sub-Experiment

Sometimes, we can get a training data set containing some basic intrusion information at a small cost. For example, the command injection attack and the response injection attack have two opposite categories in the attack mode and the data domain of the attack. It is convenient to set the corresponding classes when creating the data set. Therefore, a sub-experiment was designed to verify the optimization method when the known attack is a command injection attack or a corresponding injection attack.

The experimental data set contains seven attack classes. According to the introduction of the data set, NMRI and CMRI are response injection attacks, and MSCI, MPCI, and MFCI are command injection attacks. DoS does not directly attack the control process, but it also uses the method of sending additional commands to achieve the purpose of attack. Reconnaissance attacks use the relationship between commands and responses to perform multiple scans of the entire industrial network.

For the command injection attack, the attacker uses malicious commands to tamper with the parameters in the control process, which interferes with the normal control process and causes the abnormality of the controlled parameters. Therefore, in the command injection attack, the data of command related domain are independent variables, and the data of the response related domain are dependent variables. Since the relevance calculation method only focuses on the relevance between input changes and result changes, the method itself has no ability to distinguish between dependent variables and independent variables. Therefore, when it is known that intrusion data belong to a command injection attack, the value of the corresponding related domain can be filtered to exclude the influence of the dependent variable on the calculation result. Similarly, for response injection attacks, the data of command related domain are also interference data, which can be filtered in the calculation results.

The filtering rules above are applied to the calculation results, and the results are as follows.

For response injection attacks, two different attack types are mainly found. The relevance calculation results are mainly concentrated in [measurement: n, measurement: a] and [resp_read_fun]. Compared with the original attack classes, two attack types were found to correspond to NMRI and CMRI. All of the NMRIs corresponds to the result set [measurement: n, measurement: a], and the CMRIs corresponds to the result set [resp_read_fun]. In fact, NMRI relies on the value of malicious tampering measurement to attack the target, which is completely consistent with the filtered result. The characteristics of CMRI data are not clear. It can be known from the calculation results that the model mainly relies on resp_read_fun to detect CMRI.

For command injection attacks, the filtered results are more diverse, we sort the main results into Table 5.

It can be seen from the experiment that on the basis of the calculation results obtained by the method, it is only necessary to obtain the information that the intrusion class belongs to the command injection or the response injection, and the result with every reference value can be obtained. The analysis cost of the training set and the analysis cost of each intrusion behavior can be effectively reduced at the same time.

## 5. Discussion

In an industrial control network security system, the IDS plays a very important role. However, dealing with intrusion is a very complicated and important task for security. Our hope was to provide more useful information for subsequent analysis during the intrusion detection phase to help analysts quickly locate and solve the problems.

Deep learning is rapidly being applied to the field of intrusion detection because of its excellent continuous learning, update, and generalization capabilities. At the same time, however, providing more information beyond the output is difficult because of the lack of interpretability. Sometimes we have the ability to obtain a data set that has been processed. The data set not only knows whether the data is from intrusion, but also which intrusion class the intrusion data belongs to and the underlying characteristics of the intrusion class, such as for the data set used in the experiment. If the intrusion classification is sufficiently detailed, we can even get the information needed to solve the problem directly from the class of the data. However, the more detailed the classification, the higher the cost of the analysis, the more difficult it is for continuous learning and real-time updates. Using the reverse derivation method to obtain the classification basis information from the model can effectively solve this problem. We explored useful information that the hidden layer of a deep learning model may contain the perspective of information and the classification basis and used the LRP to display this information in an understandable way. We also designed a new data normalization method for the particulars of industrial control network data.

We have read many papers in the literature that helped us understand the deep learning model and were inspired by them. The main function of the classification model is to filter and utilize information related to the classification to finally decide the correct classification. The analysis of deep learning models from the perspective of information was highly relevant to our research and showed its relevance in our conclusions. Based on these conclusions and the working principle of the classification model, we proposed an assertion about the relationship between the model calculation process and the stepwise classification process. We designed an experiment to verify our assertion. The experimental results show that the classification bases can feasibly be extracted from the hidden layer.

The hidden layer of the deep learning model tends to converge data containing the same classification basis. The difference between the abnormal and normal data in the hidden layer can reflect the difference in classification bases. In order to eliminate the interference of irrelevant information and maintain more atomic bases, we defined and used the KDL to find comparison samples. Similar to LRP, we then passed the differences of the hidden layers to the input layer to get a result understandable by humans to help an analyst draw conclusions faster.

However, there are still many problems with the current algorithm. For example, the partial derivative calculation of the nonlinear activation function is not sufficiently accurate, and the negative values in the result are difficult to explain. The current method cannot accurately find the intrusion factor. However, the experimental results showed that this method can significantly narrow the scope of analysis and provide very valuable information.

## Figures and Tables

**Figure 1 sensors-20-03817-f001:**
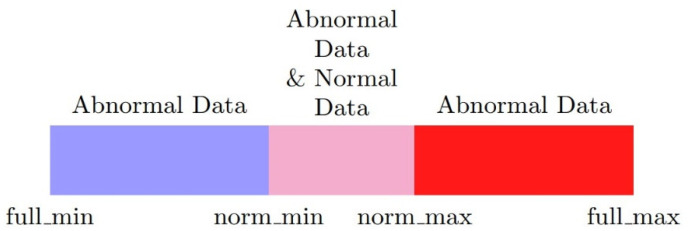
Relationship between parameters and data value fields.

**Figure 2 sensors-20-03817-f002:**
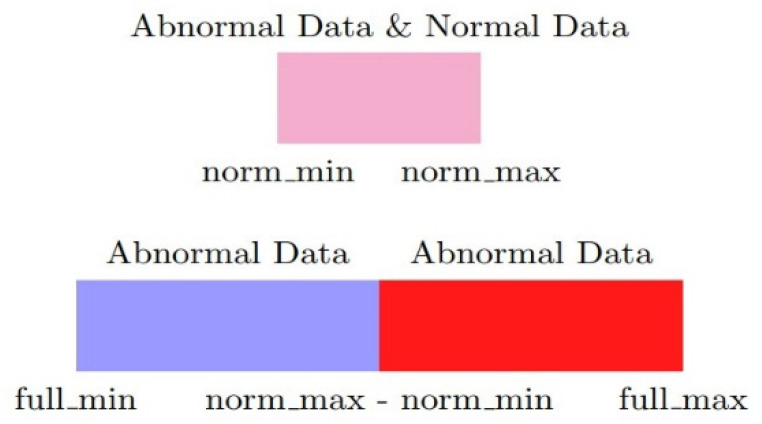
Split dimensions by parameter.

**Figure 3 sensors-20-03817-f003:**
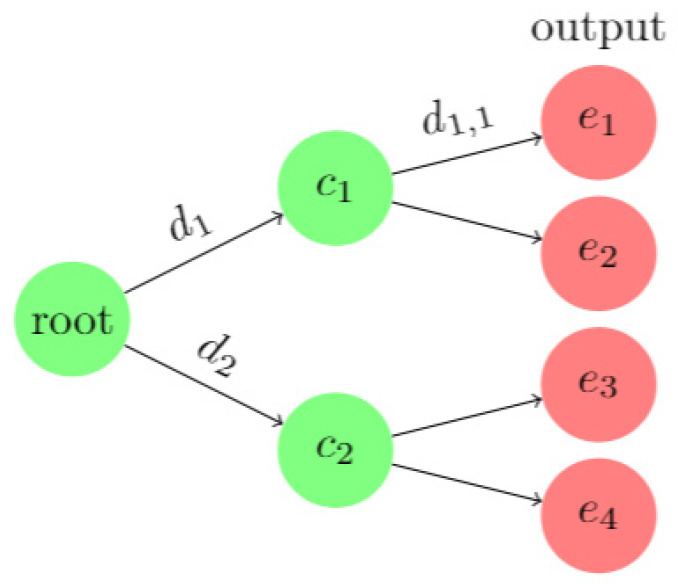
Example of a decision tree model.

**Figure 4 sensors-20-03817-f004:**
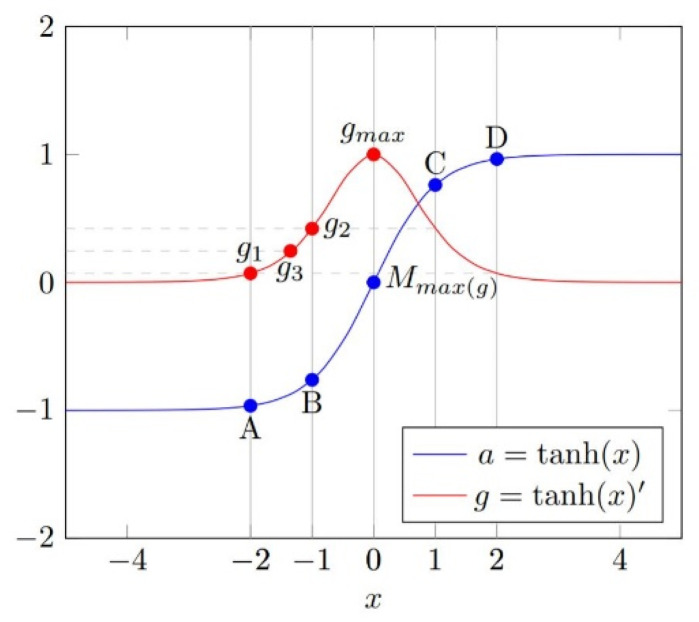
Image of tanh(x) and $\tanh{\left(x\right)}^{\prime }$.

**Figure 5 sensors-20-03817-f005:**
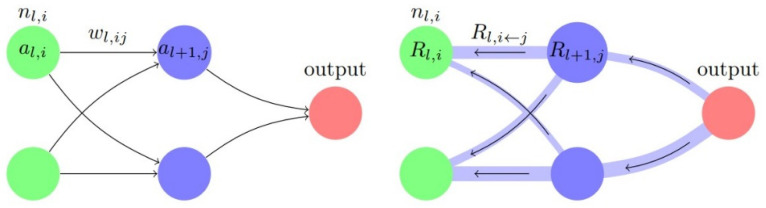
Model prediction and relevance calculation.

**Figure 6 sensors-20-03817-f006:**
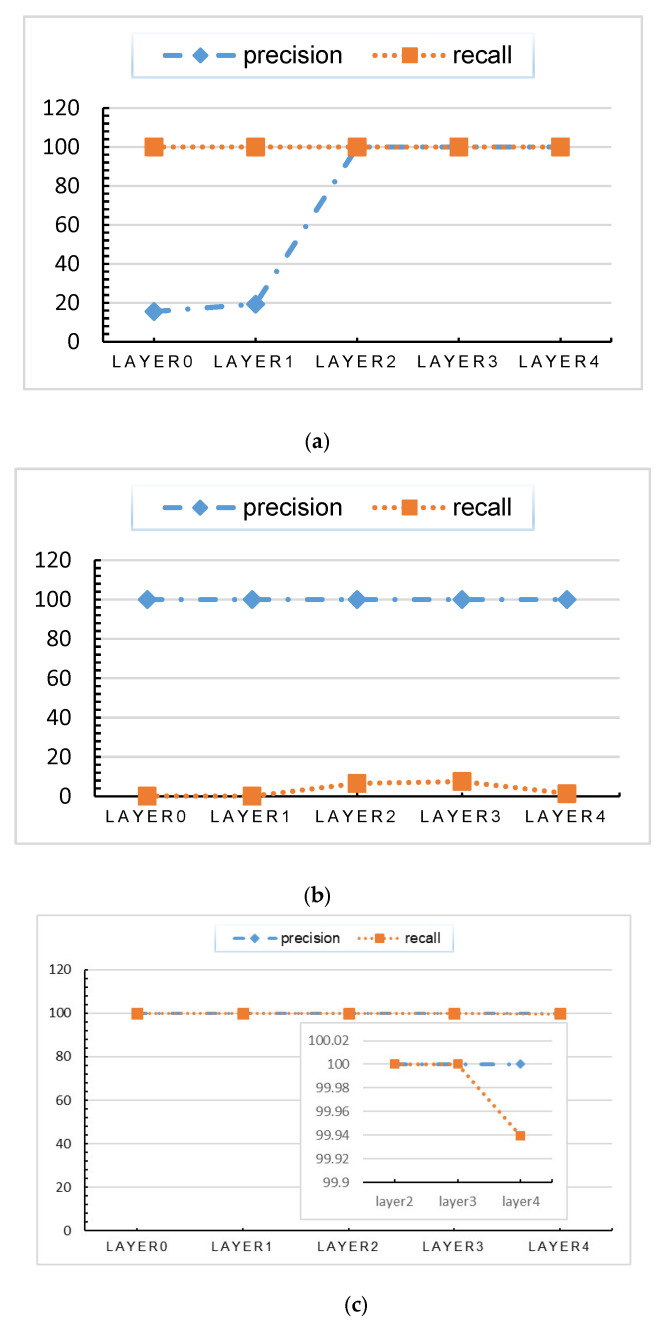
Precision and recall of the clustering results: (**a**) normal, (**b**) NMRI, (**c**) CMRI, (**d**) MSCI, (**e**) MPCI, (**f**) MFCI, (**g**) DoS, and (**h**) reconnaissance data.

**Figure 7 sensors-20-03817-f007:**
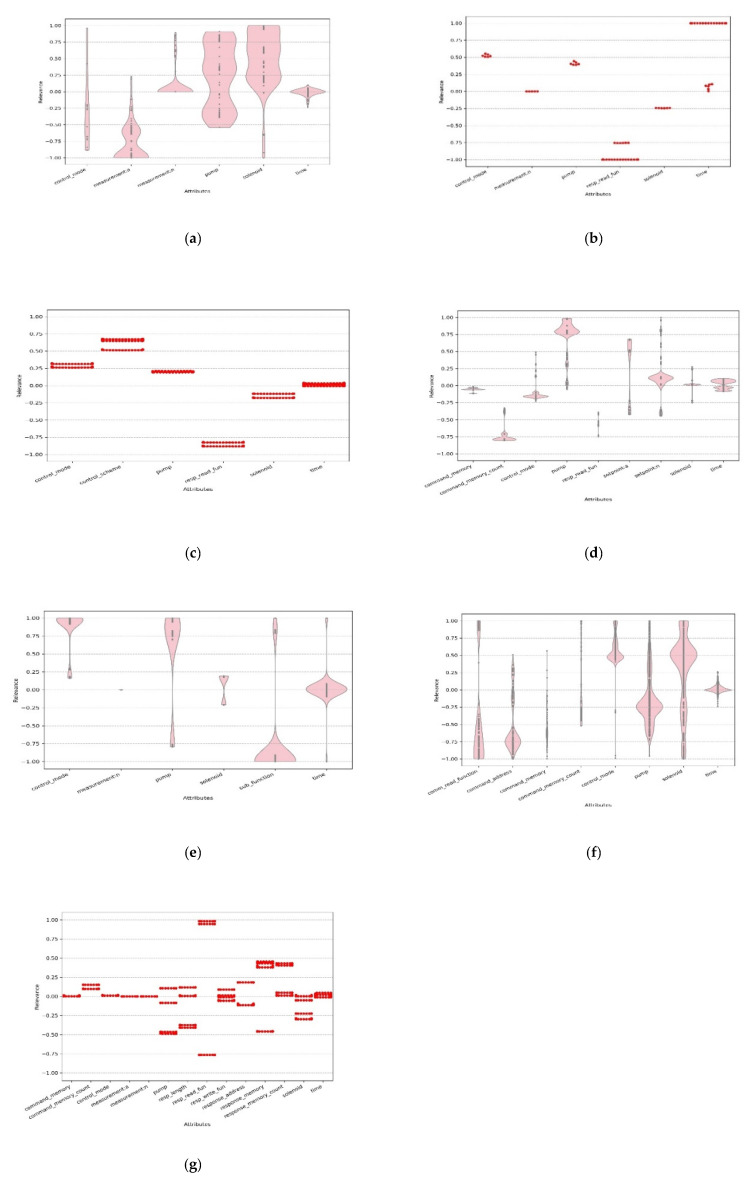
Result of relevance calculation: (**a**) NMRI, (**b**) CMRI, (**c**) MSCI, (**d**) MPCI, (**e**) MFCI, (**f**) DoS, and (**g**) reconnaissance.

**Table 1 sensors-20-03817-t001:** Precision of the clustering results of each class in each hidden layer.

Precision (%)	Layer 0	Layer 1	Layer 2	Layer 3	Layer 4
**Normal**	15.56	19.31	100	100	100
**NMRI**	100	100	100	100	100
**CMRI**	100	100	100	100	100
**MSCI**	100	100	100	100	95.74
**MPCI**	100	100	100	96.03	17.46
**MFCI**	73.17	60.98	36.59	92.68	12.50
**DoS**	61.79	63.93	65.87	19.69	1.15
**Reconnaissance**	100	100	100	100	100

**Table 2 sensors-20-03817-t002:** Recall of the clustering results of each class in each hidden layer.

Recall (%)	Layer 0	Layer 1	Layer 2	Layer 3	Layer 4
**Normal**	100	100	100	100	100
**NMRI**	0.11	0.11	6.52	7.55	1.39
**CMRI**	100	100	100	100	99.94
**MSCI**	100	100	100	100	7.49
**MPCI**	99.76	99.76	99.76	97.32	98.55
**MFCI**	5.34	92.59	93.75	36.19	16.67
**DoS**	3.19	2.95	78.30	100	11.11
**Reconnaissance**	100	100	100	100	88.37

**Table 3 sensors-20-03817-t003:** Precision of the Clustering Results for Each Class In Each Hidden Layer.

Precision (%)	Layer 0	Layer 1	Layer 2	Layer 3	Layer 4
**Normal**	100	100	88.19	87.50	100
**NMRI**	99.96	99.96	100	100	100
**CMRI**	100	100	100	100	100
**MSCI**	100	100	100	100	100
**MPCI**	61.02	61.02	100	100	100
**MFCI**	0.00	0.00	14.08	14.08	100
**DoS**	100	100	100	100	100
**Reconnaissance**	100	100	88.19	87.50	100

**Table 4 sensors-20-03817-t004:** Recall of the Clustering Results for Each Class In Each Hidden Layer.

Recall (%)	Layer 0	Layer 1	Layer 2	Layer 3	Layer 4
**Normal**	5.98	5.98	7.21	6.76	100
**NMRI**	100	100	100	100	100
**CMRI**	100	100	100	100	100
**MSCI**	100	100	100	100	100
**MPCI**	100	100	84.33	100	100
**MFCI**	0.00	0.00	94.40	94.05	100
**DoS**	100	100	100	100	100
**Reconnaissance**	5.98	5.98	7.21	6.76	100

**Table 5 sensors-20-03817-t005:** Filtered Results of Command Injection Attacks.

Attack Class	Sets of Attributes Filtered by Filtering Rules
**MSCI**	control_mode, control_scheme, solenoid, time
**MSCI**	control_mode, control_scheme, pump, solenoid, time
**MPCI**	command_memory, command_memory_count, setpoint:n, setpoint:a, control_mode, pump, time
**MPCI**	setpoint:n, setpoint:a, control_mode, pump, time
**MFCI**	sub_function, control_mode, time
**MFCI**	sub_function, pump, solenoid, time
**DoS**	command_address, control_mode, pump, solenoid, time
**DoS**	command_memory, command_memory_count, pump, solenoid, time
**DoS**	comm_read_function, pump, solenoid, time
**DoS**	comm_read_function, control_mode, pump, solenoid, time
**Reconn**	response_address, command_memory, response_memory, command_memory_count, response_memory_count, resp_write_fun, resp_length, control_mode, pump, solenoid, time

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
