# Peer review of "Explaining the Attributes of a Deep Learning Based Intrusion Detection System for Industrial Control Networks"

_sensors, 2020, doi:10.3390/s20143817_

Round 1
Reviewer 1 Report
I carefully read this work, but it is hard for me to find the contributions. Based on my understanding, the authors analyzed the DNN model and the interpretable classification model from the perspective of information, and clarify the correlation between the calculation process of the DNN model and the classification process. I don't think this topic is novel and interesting.
- The development of the attributes for DNN is not clear. The objective and motivation of this work need more enhancement.
- The contributions of the work should be summarized in Introduction.
- In Sec. 2, are the DNN models general? Can the explanations be applied to all the DNN models?
- The data normalization, hidden layer analysis, relevance calculation are the main parts of the explaining methods. I would like to see the impact of these methods on the models.
- In experiments, the authors should evaluate the improvement of the DNN models.
Author Response
- The development of the attributes for DNN is not clear. The objective and motivation of this work need more enhancement.
Response:
Sorry that we lack explanation about this. The main content of this article is the explanation of attributes and do not need to modify the model. The DNN model is the basis of the research content, so the performance of the DNN model has not been optimized. In addition, we further emphasized our objective and motivation in the second section.
- The contributions of the work should be summarized in Introduction.
Response:
Sorry that we missed an important piece. We have already made a summary about the contribution of this article at the end of the Introduction.
- In Sec. 2, are the DNN models general? Can the explanations be applied to all the DNN models?
Response:
Sorry that we lack explanation about this. The example DNN model use fully-connected layer with . In theory, models of various structures can be supported by this method, and various activation functions can also be used. In actual applications, other structures and activation functions also need to be analyzed to obtain a method that can keep the numerical relationship during relevance calculation. According to the actual design, this method will bring different levels of calculation error. When using Convolutional Neural Networks, the situation is more complicated. We have added similar descriptions to section 3.3.
- The data normalization, hidden layer analysis, relevance calculation are the main parts of the explaining methods. I would like to see the impact of these methods on the models.
Response:
Sorry that we lack explanation about this. The hidden layer analysis and relevance calculation methods are independent of the model and do not require any changes to the DNN model. The data normalization method is mainly to improve the readability of the correlation calculation results, and will have a small impact on the training of the DNN model, and the impact has been described in section 4.1.
- In experiments, the authors should evaluate the improvement of the DNN models.
Response:
Sorry that we lack explanation about this. The main goal of this article is to explain the attributes of the intrusion data and try to make the method applicable to a wider range of requirements, so improving the performance of the DNN model is not the main task.
Reviewer 2 Report
The authors address the process of intrusion detection using Deep Learning (DL) which is an interesting area. The aim is to understand how Deep Learning comes up with the decision and extract information from its hidden layers using a similar concept of layer-wise relevance propagation (LRP). The authors try to find the attribute and the layer used by the model to make the key decision during the classification task. Also, they proposed a new normalization method to improve the discrimination of the input data and compare its performance against the traditional normalization method. The novelty of the paper is high as there is a pressing need to improve the transparency and interpretability of DL.
However,
the paper would benefit from a related work section in the context of intrusion detection systems (20 references is not enough, please reach 40 at least). The authors should add a related work section
More information about the dataset used in this paper is required as well.
The authors should explain the choice of the activation function (TanH) and why they did not test with other activation functions.
My final question is about scalability: can the computation work efficiently with a big dataset or with a more complex model.
The availability of the source code and the dataset is appreciated (even if it normal, it is not totally current)
Please revise carefully the layout of the paper:
Figure7 w/o caption on page 16
The table 5 is cutted
And do a carefull proof reading
Minors:
plane 17 (page 5)
is large is proposed (page 17)
xAis page 8
Gao1818? please give a reference there
what ICS means (page 10)?
provide an acronym summary table on section 3.2.1
Author Response
- The authors should add a related work section.
Response:
Sorry that we missed an important piece. We have added related works section to the article.
- More information about the dataset used in this paper is required as well
Response:
Sorry that we missed an important piece. We have added detailed information about the dataset to the section 4.1.1.
- The authors should explain the choice of the activation function (TanH) and why they did not test with other activation functions.
Response:
Sorry that we lack explanation about this. In method section, the example DNN model use fully-connected layer with . In theory, models of various structures can be supported by this method, and various activation functions can also be used. In actual applications, other structures and activation functions also need to be analyzed to obtain a method that can keep the numerical relationship during relevance calculation. According to the actual design, this method will bring different levels of calculation error. If is used, the situation is very similar to , and the situation is different when using , when using Convolutional Neural Networks (CNN), the situation is more complicated. We have added similar descriptions to section 3.3.
- Can the computation work efficiently with a big dataset or with a more complex model?
Response:
Sorry that we lack explanation about this. This method will take longer to find a comparison sample when facing a larger data set, but the time complexity is , and the time of single calculation is also very short, which will not cause excessive time overhead. In the relevance calculation method, intermediate results such as partial derivative numerical calculation can be cached to improve the performance of the method. Since some approximation methods are used in the derivation of the formula, the calculation error is inevitable, so the calculation error will be greater when the model is larger. However, it can be found through experiments that this method can complete the work well in a small-scale model under the ISC environment. We have added similar descriptions to section 3.4.
Reviewer 3 Report
This is a well-written paper and the authors have provided an interesting solution for intrusion detection in industrial networks. However, there are some comments that should be addressed to improve the paper.
- While this paper is about intrusion detection in industrial networks, there was not enough literature review on the security of industrial networks.
- More information is needed on the dataset used for the evaluation of the proposed methods. Is this a dataset that is generated in this study? More details about the implemented industrial testbed should be provided.
- The authors have not compared their results with other methods or other studies.
Author Response
- While this paper is about intrusion detection in industrial networks, there was not enough literature review on the security of industrial networks.
Response:
Sorry that we missed an important piece. We have added related works section to the article.
- More information is needed on the dataset used for the evaluation of the proposed methods. Is this a dataset that is generated in this study? More details about the implemented industrial testbed should be provided.
Response:
Sorry that we missed an important piece. We have added detailed information about the dataset to the section 4.1.1.
- The authors have not compared their results with other methods or other studies.
Response:
Sorry that we missed an important piece. For the intrusion detection part, we did not spend too much work on it, and used simple structure models in experiments, resulting in lower accuracy than many studies focused on improving accuracy, but the difference is not much. For the correlation calculation part, because there are not many related studies and the lack of comparison standards, no comparison test was conducted.
Round 2
Reviewer 1 Report
My comments have been addressed.
Reviewer 2 Report
I am satisfied with the author's reply
(please consider in the related works section to cite the author and give the key e.g Carano et. [12])